# Is Education Alone Enough to Sustain Improvements of Antimicrobial Stewardship in General Practice in Australia? Results of an Intervention Follow-Up Study

**DOI:** 10.3390/antibiotics12030594

**Published:** 2023-03-16

**Authors:** Robin Sangwan, Alicia J. Neels, Stella May Gwini, Sajal K. Saha, Eugene Athan

**Affiliations:** 1School of Medicine, Faculty of Health, Deakin University, Geelong, VIC 3220, Australia; 2Research Department, University Hospital Geelong, Barwon Health, Geelong, VIC 3220, Australia; 3Centre for Innovation in Infectious Disease and Immunology Research (CIIDIR), Barwon Health, Geelong, VIC 3220, Australia; 4Institute for Mental and Physical Health and Clinical Translation (IMPACT), Deakin University, Geelong, VIC 3000, Australia; 5National Centre for Antimicrobial Stewardship, The University of Melbourne, Melbourne, VIC 3000, Australia

**Keywords:** antimicrobial stewardship, education programme, follow-up study, general practice, primary care

## Abstract

Sustained behaviour change and practice improvements for the optimal use of antimicrobials remains challenging in primary care. In 2018, a simple antimicrobial stewardship education programme involving guideline recommendations for common infections, antimicrobial audit reports, and local antibiograms resulted in significant improvements in guideline compliance and more appropriate antimicrobial prescribing by GPs. This observational follow-up study aims to examine the sustainability of the positive intervention effect after two years of implementation of the intervention. Practice-based data on all oral antimicrobial prescriptions issued by GPs were collected retrospectively to compare with intervention data and to measure the sustainability of the intervention effect. The data were analysed using a two-sample test of proportions. The primary outcomes included changes in the rate of prescription compliance with the Australian “Therapeutic Guidelines: Antibiotic” and the appropriateness of antimicrobial choice and duration of therapy. Overall, there was a significant decline in guideline compliance, from 58.5 to 36.5% (risk ratio (RR) (95% CI): 0.62 (0.52–0.74)), in the appropriateness of antimicrobial choice, from 92.8 to 72.8% (0.78 (0.73, 0.84)), and in the prescribed duration, from 87.7 to 53.3% (0.61 (0.54, 0.68)) in the intervention follow-up period. In respiratory infections and ear, nose, and throat infections, the rates of guideline compliance and appropriate choice and duration of antimicrobial prescription decreased significantly at *p* < 0.001. Appropriateness in the duration of antimicrobial therapy also significantly decreased for most antimicrobials. The evidence suggests that a simple and single-occasion antimicrobial stewardship education programme is probably not enough to sustain improvements in the optimal use of antimicrobials by GPs. Future research is needed to validate the results in multiple GP clinics and to examine the effect of sustained education programmes involving infection-specific and antimicrobial-targeted audits and feedback.

## 1. Introduction

The overuse and inappropriate use of antibiotics substantially contributes to growing antimicrobial resistance (AMR) [1]. In Australia, antibiotics continue to be overprescribed in primary care; 81.5% of patients with acute bronchitis and 80.1% of patients with acute sinusitis were prescribed antimicrobials, though there is no evidence of benefit [2]. General practitioners (GPs) in Australia prescribe antibiotics up to nine times more frequently than therapeutic guidelines allow for, particularly for respiratory infections [3].

Increasing evidence suggests that antimicrobial stewardship (AMS) education interventions can improve the appropriate prescription of antimicrobials by GPs [4,5,6]. Interventions include education sessions, decision support tools, and online modules [4,5,6]. Though AMS education has shown success in reducing antibiotic prescriptions [7,8,9], less attention has been paid to examining the sustainability of these effective interventions [10]. The sustained behavioural change at the clinician and patient level is a pivotal aspect to the safe use of antimicrobials.

Neels et al. (2020) [8] undertook a simple AMS education intervention in 2018 in a large general practice clinic in regional Victoria, Australia, with an aim to optimise antimicrobial prescribing by GPs. The specific components of their intervention included face-to-face education sessions with GPs that emphasised AMS principles, antimicrobial resistance, current prescribing guidelines, and microbiological testing. A one-hour face-to-face academic detailing session was implemented with GPs. The results of the initial audits of antimicrobial prescription, guideline recommendations for common infections, regional antibiograms and patterns of resistance, and AMS techniques such as delayed prescribing were shared. An infectious disease physician and AMS pharmacist facilitated the education session.

Neels et al. (2020) [8] found significant post-intervention improvements in terms of guideline compliance and the appropriateness of antimicrobial prescribing practices by GPs; however, a valid question remains as to whether the improvements persist. In the current literature, evidence that demonstrates the sustainability of effective AMS interventions in primary care is extremely limited. This evidence gap is a significant barrier for policy proposition around the implementation of AMS programmes in primary care. Therefore, the primary objective of this follow-up study was to examine whether the improvements in guideline compliance and appropriateness in terms of the choice and duration of antimicrobial prescription(s) were sustained in 2019.

## 2. Results

### 2.1. Patient and Prescription Demographics

The demographics of the patients and prescriptions are presented in Table 1. In total, 368 patients were prescribed at least one antimicrobial in the post-intervention period compared to 351 patients in the follow-up period. The mean age (years) of the patients prescribed antimicrobials was higher among the follow-up cohorts compared to post-intervention (47.8 vs. 43.1). No statistically significant differences were found in the age or gender distributions between the two cohorts. The number of antimicrobial prescriptions reviewed was 373 and 336 in the post-intervention and follow-up periods, respectively, with no significant differences (*p* < 0.052).

### 2.2. Compliance with Guidelines

Table 2 shows the rate of guideline compliance in the prescription of antimicrobials. Overall, there were 22% significant reductions in the guideline compliance rate from 58.5% in 2018 to 36.5% in 2019 (risk ratio (RR) (95% CI): 0.62 (0.52–0.74)).

Table 3 presents the changes in guideline compliance according to the antimicrobial drugs. The rate of guideline compliance dropped significantly for doxycycline by 36.5% (from 83.9% (52/62) to 52.6% (27/57), *p* < 0.001), amoxicillin by 47.5% (from 76.7% (46/60) to 29.2% (14/48), *p* < 0.0001), and cefalexin by 16.9% (from 31.4% (16/51) to 14.5% (8/55), *p* = 0.039). Nonsignificant reductions in guideline compliance were observed for amoxicillin with clavulanic acid, flucloxacillin, phenoxymethylpenicillin, clarithromycin. In contrast, guideline compliance improved for trimethoprim by 5.8%, metronidazole by 16.7%, and trimethoprim with sulfamethoxazole by 50%, but without achieving statistical significance (Table 3).

The antibiotic prescription guideline compliance significantly reduced in patients with respiratory infections by 43.9% (67.8 to 23.9%, *p* < 0.001) and ear, nose, and throat infections by 29.7% (49.3 to 19.6%, *p* = 0.001) (Table 4). Nonsignificant reductions in guideline compliance were found in gastrointestinal tract infections of 25.5% (58.8 to 33.3%), skin and soft tissue infections (including acne) of 16.1% (54.7 to 38.6%), and medical prophylaxis of 24.2% (84.2 to 60%) (Table 4). In contrast, guideline compliance improved by 1.5% (44.7 to 46.2%) in urinary tract infections and by 5.6% (72.7 to 78.3%) in genital and sexually transmitted infections, but without reaching statistical significance (Table 4).

### 2.3. Appropriateness of Prescription by Choice of Antimicrobial Drug

Overall, the appropriate selection of antimicrobial drugs significantly declined by 20% from 92.8% (post-intervention period) to 73.8% (follow-up period) (RR (95% CI):0.78 (0.73, 0.84) (Table 2).

Table 3 depicts the frequency of the prescription of individual antimicrobial drugs. The most frequently prescribed antibiotic was amoxicillin, accounting for nearly one-fifth of the antimicrobial prescriptions during both the intervention and follow-up periods, followed by doxycycline (17.7 and 19.4%), cefalexin (15.3 and 17.6%), and amoxicillin/clavulanic acid (13.1 and 9.8%). While the proportion of prescriptions increased for some individual antibiotics in 2019 (e.g., doxycycline, cefalexin, azithromycin, phenoxymethylpenicillin), and decreased for others (e.g., amoxicillin, metronidazole, flucloxacillin), the difference in the prescription pattern was only statistically significant for azithromycin (an increase from 1.1% in 2018 to 3.3% in 2019, *p* < 0.042) (Table 3).

Appropriateness significantly declined for amoxicillin (38.2%, *p* < 0.001), amoxicillin/clavulanic acid (25.6%, *p* < 0.023), cefalexin (21.3%, *p* < 0.001), and clarithromycin (80%, *p* < 0.007) in the follow-up period. Appropriateness remained unchanged for flucloxacillin, azithromycin, tinidazole, and trimethoprim. Improved appropriateness was observed in the selection of phenoxymethylpenicillin and trimethoprim with sulfamethoxazole, but without reaching statistical significance (*p* > 0.999) (Table 3).

Respiratory infections and ear, nose, and throat infections were the most common clinical indications where there were significant reductions in the appropriateness of choosing antimicrobials. In respiratory infections, appropriateness reduced by 44.9% (87.0 to 42.1%, *p* < 0.001) and in ear, nose, and throat infections by 20.2% (94.7 to 74.5%, *p* = 0.001) (Table 4). No infections were found where appropriateness remained unchanged or improved in the follow-up period.

### 2.4. Appropriateness of Prescription by the Duration of Antimicrobial Therapy

Overall, the rate of appropriate duration of prescribed antimicrobial therapy reduced by 35% from 87.7 to 53.3% (RR (95% CI): 0.61 (0.54, 0.68)) (Table 2). The rate of appropriate duration significantly declined in the follow-up period for six antimicrobials: amoxicillin (40%, *p* < 0.001), amoxicillin/clavulanic acid (52.2%, *p* < 0.001), cefalexin (48%, *p* < 0.001), doxycycline (25.6%, *p* < 0.002), clarithromycin (60%, *p* < 0.035), roxithromycin (100%, *p* < 0.003), and erythromycin (71.4%, *p* < 0.021). The appropriate duration remained unchanged for azithromycin and tinidazole (Table 3).

The appropriateness significantly decreased in the duration of prescribed antimicrobials in five common infections: respiratory infections by 58.2% (*p* < 0.001), gastrointestinal tract infections by 50.9% (*p* = 0.002), skin and soft tissue infections (including acne) by 26.4% (*p* < 0.001), ear, nose, and throat infections by 35% (*p* < 0.001), and urinary tract infections by 18.3% (*p* = 0.050) (Table 4). The appropriateness of the rate of duration did not improve or stay the same for any of the other infections during the follow-up period.

## 3. Discussion

This intervention follow-up study of Neels et al. [8] examined whether the post-intervention improvements of antimicrobial prescribing by GPs were sustained in the follow-up period across the major outcome measures of guideline compliance, appropriate choice, and duration of course. Overall, guideline compliance, appropriateness of antimicrobial choice, and appropriateness of the duration of antimicrobial therapy significantly declined by 22, 20, and 35%, respectively (Table 2).

Our findings can be compared with a systematic review concluding that the effect of AMS interventions utilising clinical practice guidelines, audits and feedback, and educational materials lasts in the short to medium term [9]. García-Rodríguez et al. conducted consistent educational interventions for seven years and found a significant improvement in the rate of appropriateness, with 50% improvements found for amoxicillin with clavulanic acid [11]. One year after the intervention, our study found nonsignificant reductions in guideline compliance of 12%, and significant reductions in appropriate choice of 25.6% and appropriate duration of 47% for amoxicillin with clavulanic acid. Similarly, both appropriateness and guideline compliance were significantly reduced for the broad-spectrum antibiotics cefalexin and clarithromycin. These results indicate that future education and feedback programmes can be targeted to broad-spectrum antibiotics.

We found mixed results in terms of the changes in guideline compliance according to antimicrobials and indications. Guideline compliance was significantly reduced during the follow-up period for amoxicillin, doxycycline, and cefalexin antibiotics and for respiratory and ear, nose, and throat infections. In contrast, prescriptions for trimethoprim, metronidazole, and trimethoprim with sulfamethoxazole showed improved guideline compliance. In urinary tract infections and genital and sexually transmitted infections, there were signs of improved guideline-compliant antimicrobial use. These findings emphasise that GPs need more directed and repeated intervention support to deal with common respiratory and ear, nose, and throat infections and commonly used antibiotics.

Limiting courses of antimicrobials to their most appropriate duration is an incredibly important aspect of AMS programmes to reduce patient adverse effects and prevent the development of resistance [1,12]. In our study, improvements in the appropriate duration of antimicrobial therapy significantly decreased for all antimicrobials except for azithromycin, tinidazole, and trimethoprim. There were no infections for which the improved appropriateness of duration was sustained or improved further. The duration of antimicrobial therapy should be an important focus in future AMS education and feedback programmes, and qualitative study exploring the reasons for improving the appropriateness in duration is worth trying.

Interestingly, GPs maintained appropriate duration when prescribing trimethoprim. For cystitis in non-pregnant women, trimethoprim was often prescribed as a full pack containing 7 days of therapy, though the guidelines recommended 3 days [13]. Following the intervention, guideline compliance for trimethoprim almost doubled and improved even more in this follow-up study. This finding emphasises that Australian antibiotic pack sizes should be aligned with the recommended treatment guidelines to partly address duration problems. The Therapeutic Goods Administration (TGA) of the Department of Health in Australia should work with manufacturers to align antibiotic pack sizes with indications where there is international consensus on dose and treatment duration. As of 2021, the cautionary advisory label ‘Take for the number of days advised by your prescriber’ is provided by pharmacists in Australia on antibiotic packaging and patients are advised to take any ‘leftover’ antibiotics back to the pharmacy for disposal, which could have a positive impact on optimising the course duration [14].

Failure to sustain or improve intervention effects can be multifactorial. As data were collected and analysed against the new eTG guideline “Antibiotic” version 16, there may have been variation [15] in the guidelines that the GPs were not familiar with. The new guidelines were only available for 3 months prior to the month of data collection, July 2019. This fact can be supported by the finding that GPs in Australia prescribed a number of antibiotics that had been removed from the Therapeutic Guidelines recommendation list [16]. In our post-intervention study [8], the GPs acknowledged that they had a lack of access to the online version of the guidelines. The other factors that prevent GPs from following optimal prescribing behaviours include patient pressure, patient expectations [17], and repeat prescriptions [18]. Repeat prescriptions are common in general practice in Australia [19] and often lead to a duration of therapy that is longer than the guidelines recommend [11]. A no repeat prescription policy [20] was only introduced in 2020; thus, this issue may have contributed to the obtained results.

The lessons learned from this study indicate that AMS interventions should not be a one-time project, but instead be an ongoing multifaceted programme [21,22,23] that can provide avenues of support for when GPs are unsure about certain antibiotic prescribing via peer-to-peer networks with pharmacists, and with semi-regular medication reviews and guideline updates. Undertaking system thinking approaches [24,25] to establish practice-based AMS programmes with GP training could be a more sustainable approach, but this requires future research and development. A holistic data-driven approach [25] could be considered by policymakers for the sustained improvement and delivery of targeted stewardship education programmes. In addition to practice-level prescribing data, the local, regional, and national AMR data in primary care needs to be generated and aggregated to continually develop and update local antibiotic guidelines and decision support tools in an iterative process, which could support GPs and pharmacists to promote evidence-based prescribing. The establishment of GP–pharmacist quality circles, as evidenced in Switzerland [26,27], could potentially sustain improvement activities related to optimal antimicrobial use in Australia. Quality circles provide an avenue for the GP–pharmacist interprofessional sharing of evidence-based recommendations, setting goals and strategies for prescribing targets for particular infections, tackling patient expectations and peer feedback on antimicrobial audits, and identifying new knowledge requirements, and the programme itself increases the feeling of ownership for change. Ratajczak et al. [28] found that repeated social norm feedback confirmed by a consistent and continuous programme could have greater potential in promoting safe antimicrobial prescribing behaviours in primary care. Globally, implementation research on developing antimicrobial stewardship in primary care is limited [29], and further study in this area would, therefore, contribute to direct future implementation research and approaches in Australian primary care.

### Strengths and Limitations

This is the first Australian study measuring the sustainability of the observed benefits of education and training on the prescribing behaviour of GPs. A major strength of this study is that it reviewed GP consultations directly by looking at GP consultation notes, which included clinical and observational notes, recorded indications, and any patient-to-GP communication that occurred during the consultation, whereas other recent studies primarily utilised antibiotic dispensing data via the PBS [20]. This cohesive methodology provides investigators with a more pragmatic viewpoint on prescribing habits, such as compliance or prescription issues concerning antibiotics.

There were some limitations in our study. This single-centred study may not be representative of other GP clinics and, thus, the results may not be generalisable in other states or countries. A lack of adequate clinical notes caused some prescriptions to be excluded from the analysis. Our data did not consider how many GPs dissuaded patients from the use of antibiotics and the consultations where GPs did not prescribe antibiotics despite a patient having an infection. Future research may identify the proportion of antibiotic prescriptions denied by GPs. When no antibiotic duration was specified within a prescription, the total amount provided was considered the antibiotic course duration. This could have been communicated to the patient, but not documented in the GP notes. The data were analysed against the new eTG guidelines (“Antibiotic” version 16), and there may have been variations [15] in the guidelines that the GPs were unfamiliar with. This study did not count the Hawthorne effect during statistical analysis. The rates of guideline compliance and appropriateness might be influenced by the fact that GPs who attended the educational intervention in 2018 may have left the clinic or been absent from work for the month of data collection in 2019. Finally, only one month of data from 2018 and one from 2019 were collected and compared, and a more extended data collection period could provide varied results.

## 4. Conclusions

The effects of a simple and single-occasion AMS education programme declined over time in terms of improving guideline compliance and appropriateness of antimicrobial prescription in general practice. Future studies are required to validate this result with multiple GP clinics, and to determine whether long-term effects can be achieved with a sustained AMS programme involving antibiotic guideline updates and infection-directed and antimicrobial-targeted audits and feedback programmes.

## 5. Methods

This was a retrospective observational follow-up study. We collected practice-based data on all oral antimicrobial prescriptions issued by GPs in July 2019. The intervention rolled out to GPs in June 2018 consisted of a one-hour face-to-face academic detailing session on appropriate antibiotic prescribing for common presentations, regional antibiogram, resistance patterns, and delayed prescribing. Hard copies of the Australian “Therapeutic Guidelines: Antibiotic” version 15 were supplied to GPs.

The post-intervention data (July 2018) were compared with the intervention follow-up data (July 2019) to observe the difference and measure the sustainability of the intervention effects. The data were collected using the same method as described in the published study by Neels et al. [8] to allow for comparisons between the two-time points. REDcap, a secure web platform, were used for data management.

The data collected on antimicrobial use included whether an antimicrobial was prescribed, the name of the antimicrobial prescribed, the dosage (e.g., 50 mg tablet), the duration (e.g., 1 tablet daily for 5 days), and the indication for prescribing the antimicrobial(s). Patient demographic information and all clinical observations, notes taken during consultations, microbiological or radiological requests, and consultation outcomes were recorded to aid in the judgement of guideline compliance and appropriateness of the prescription. Any prescription with missing information to assess appropriateness was deleted from the analysis.

A prescription assessment algorithm based on “Therapeutic Guidelines: Antibiotic” version 16 [13] was used for the assessment of guideline compliance and appropriateness. An infectious disease physician (EA) and antimicrobial stewardship pharmacist (AN) assessed the appropriateness and guideline compliance. A prescription was deemed compliant with the guideline when the choice, dosage, frequency, and duration all aligned with the guideline. A prescription was considered appropriate when the antibiotic choice, dosage, and duration were a reasonable selection for the indication provided based on a set of criteria that included a spectrum of activity, pharmacokinetics of the antibiotic, and the disease itself. Team-based assessment was conducted to avoid any bias in the results.

The Stata Statistical Software BE 17^®^ (StataCorp, 2015. College Station, TX, USA: StataCorp LP) was used for the analysis. The rate of guideline compliance, appropriateness of antimicrobial therapy, and other variables were determined using descriptive statistics. Categorical data were summarised through the usage of frequencies and percentages. A chi-square test was used to compare the demographic variables. The comparison of the guideline compliance and appropriateness between the 2018 and 2019 cohorts was carried out using Poisson regression with robust sandwich error estimates, and the results were reported as risk ratios (RR) with 95% confidence intervals (CI). A difference was considered statistically significant if *p* < 0.05. The Strengthening the Reporting of Observational Studies in Epidemiology (STROBE) guideline was used to report this study (Appendix A).

## Figures and Tables

**Table 1 antibiotics-12-00594-t001:** Descriptive statistics describing patient and prescription demographics of the study in Australia.

	July 2018	July 2019
Patients prescribed antibiotics	386	351
Patients prescribed antibiotics who were excluded from analysis	17	29
Number of patients included in analysis	369	322
Average patient age at antibiotic prescription (years)	43.1 (SD = 25.7)	47.8 (SD = 23.5)
Median patient age at antibiotic prescription (years) (IQR)	42 (24, 65)	48 (29, 69)
Number of female patients prescribed antibiotics (%)	252 (67. 6%)	240 (71.4%)
Total antibiotic prescriptions reviewed	373	336
1 antibiotic prescribed	365	308
2 antibiotics prescribed	4	14
Total number of prescriptions eligible for determining guideline compliance	316	301
Total number of prescriptions eligible for determining appropriateness of choice of antimicrobials	362	305
Total number of prescriptions eligible for determining the appropriateness of prescribed duration	360	304

IQR = interquartile range (25th percentile, 75th percentile).

**Table 2 antibiotics-12-00594-t002:** Changes in guideline compliance and appropriateness of choice and duration of antimicrobial prescription.

	July 2018	July 2019	Change
	n/N (%)	n/N (%)	RR (95% CI) (*p*-Value)
Compliance with guidelines	185/316 (58.5)	110/301 (36.5)	0.62 (0.52–0.74) (*p* < 0.001)
Appropriateness			
Antimicrobial (choice)	336/362 (92.8)	222/305 (72.8)	0.78 (0.73, 0.84) (*p* < 0.001)
Duration of prescribed antimicrobial(s)	316/360 (87.7)	162/304 (53.3)	0.61 (0.54, 0.68) (*p* < 0.001)

Note: n = number of prescriptions compliant with/appropriate according to the Australian Therapeutic Guidelines. N = total number of prescriptions.

**Table 3 antibiotics-12-00594-t003:** Comparison of appropriateness and guideline compliance of antimicrobial prescriptions by antimicrobial type.

Antimicrobial	Total Number of Prescriptions for Each Antibiotic (Including Prescriptions Ineligible for Compliance and/or Appropriateness Assessment)		Guideline Compliance (Only Eligible Prescriptions Included) n/N(%)			Appropriateness (Only Eligible Prescriptions Included)					
						Of Prescription Choice, n/N(%)			Of Prescription Duration, n/N(%)		
	2018	2019	2018	2019	*p*-Value	2018	2019	*p*-Value	2018	2019	*p*-Value
Amoxicillin	71 (19.0)	55 (16.4)	46/60 (76.7)	14/48 (29.2)	<0.001	68/69 (98.6)	29/48 (60.4)	<0.001	65/69 (94.2)	26/48 (54.2)	<0.001
Amoxicillin with clavulanic acid	49 (13.1)	33 (9.8)	15/42 (35.7)	7/30 (23.3)	0.261	34/47 (72.3)	14/30 (46.7)	0.023	44/47 (93.6)	12/29 (41.4)	<0.001
Azithromycin	4 (1.1)	11 (3.3)	2/2 (100.0)	10/10 (100.0)	---	4/4 (100.0)	11/11 (100.0)	---	4/4 (100.0)	11/11 (100.0)	---
Cefaclor	6 (1.6)	2 (0.6)	0/5 (0.0)	0/2 (0.0)	---	3/5 (60.0)	0/2 (0.0)	0.429	5/5 (100.0)	0/2 (0.0)	0.048
Cefalexin	57 (15.3)	59 (17.6)	16/51 (31.4)	8/55 (14.5)	0.039	53/54 (98.1)	43/56 (76.8)	0.001	49/55 (89.1)	23/56 (41.1)	<0.001
Clarithromycin	8 (2.1)	5 (1.5)	3/7 (42.9)	0/5 (0.0)	0.205	8/8 (100.0)	1/5 (20.0)	0.007	8/8 (100.0)	2/5 (40.0)	0.035
Doxycycline	66 (17.7)	65 (19.4)	52/62 (83.9)	27/57 (47.4)	<0.001	63/64 (98.4)	46/57 (80.7)	0.001	52/61 (85.2)	34/57 (59.6)	0.002
Erythromycin	7 (1.9)	7 (2.1)	0/6 (0.0)	0/7 (0.0)	---	5/7 (71.4)	1/7 (14.3)	0.103	5/7 (71.4)	0/7 (0.0)	0.021
Flucloxacillin	10 (2.7)	6 (1.8)	7/10 (70.0)	1/5 (20.0)	0.119	10/10 (100.0)	6/6 (100.0)	---	9/10 (90.0)	4/6 (66.7)	0.518
Metronidazole	27 (7.2)	18 (5.4)	6/12 (50.0)	10/15 (66.7)	0.381	26/27 (96.3)	13/15 (86.7)	0.287	23/27 (85.2)	10/15 (66.7)	0.242
Phenoxymethylpenicillin	8 (2.1)	13 (3.8)	2/6 (33.3)	2/11 (18.2)	0.584	6/8 (75.0)	9/11 (81.8)	>0.999	5/8 (62.5)	5/11 (45.5)	0.650
Roxithromycin	4 (1.1)	7 (2.1)	1/4 (25.0)	0/7 (0.0)	0.364	2/4 (50.0)	0/7 (0.0)	0.109	4/4 (100.0)	0/7 (0.0)	0.003
Tinidazole	3 (0.8)	3 (0.9)	2/2 (100.0)	3/3 (100.0)	---	3/3 (100.0)	3/3 (100.0)	---	3 (100.0)	3 (100.0)	---
Trimethoprim	26 (7.0)	25 (7.4)	14/23 (60.9)	16/24 (66.7)	0.679	26/26 (100.0)	24/24 (100.0)	---	18/26 (69.2)	17/24 (70.8)	0.902
Trimethoprim with sulfamethoxazole	10 (2.7)	2 (0.6)	4/8 (50.0)	2/2 (100.0)	0.467	9/10 (90.0)	2/2 (100.0)	>0.999	8/10 (80.0)	2/2 (100.0)	>0.999
Other **	19 (5.1)	22 (6.5)									

Note: n = number of prescriptions for that drug that were compliant with or had appropriate drug choice/duration according to the Australian Therapeutic Guidelines. N = total number of prescriptions for that drug that were eligible to be assessed for compliance or appropriateness. ** includes cefuroxime, ciprofloxacin, clindamycin, dicloxacillin, fluconazole, hexamine hippurate, mefloquine, minocycline, nitrofurantoin, norfloxacin, oseltamivir, rifampicin.

**Table 4 antibiotics-12-00594-t004:** Comparison of appropriateness and guideline compliance of antimicrobial prescription by type of clinical indication.

Antimicrobial	Total Number of Prescriptions within Each Indication (Including Prescriptions Ineligible for Compliance and/or Appropriateness Assessment)		Guideline Compliance (Only Eligible Prescriptions Included) n/N(%)			Appropriateness (Only Eligible Prescriptions Included)					
						Of Prescription Choice, n/N(%)			Of Prescription Duration, n/N(%)		
	2018, n(%)	2019, n(%)	2018	2019	*p*-Value	2018	2019	*p*-Value	2018	2019	*p*-Value
Ear, nose and throat infections	76 (20.4)	49 (14.6)	33/67 (49.3)	9/46 (19.6)	0.001	71/75 (94.7)	35/47 (74.5)	0.001	67/76 (88.2)	25/47 (53.2)	<0.001
Gastrointestinal tract infections	20 (5.4)	16 (4.8)	10/17 (58.8)	5/15 (33.3)	0.149	17/19 (89.5)	9/15 (60.0)	0.100	16/19 (84.2)	5/15 (33.3)	0.002
Genital and sexually transmitted infections	28 (7.5)	27 (8.0)	8/11 (72.7)	18/23 (78.3)	>0.999	28/28 (100.0)	23/24 (95.8)	0.462	24/28 (85.7)	20/24 (83.3)	>0.999
Prophylaxis: medical	25 (6.7)	27 (8.0)	16/19 (84.2)	6/10 (60.0)	0.148	20/21 (95.2)	9/11 (81.8)	0.266	16/18 (88.9)	8/11 (72.7)	0.339
Respiratory infections	95 (25.5)	94 (28.0)	59/87 (67.8)	21/88 (23.9)	<0.001	80/92 (87.0)	37/88 (42.1)	<0.001	87/92 (94.6)	32/88 (36.4)	<0.001
Skin and soft tissue infections (including acne)	69 (18.5)	59 (17.6)	35/64 (54.7)	22/57 (38.6)	0.077	63/69 (91.3)	50/58 (86.2)	0.361	63/69 (91.3)	37/57 (64.9)	<0.001
Urinary tract	54 (14.5)	53 (15.8)	21/47 (44.7)	24/52 (46.2)	0.883	54/54 (100.0)	52/52 (100.0)	---	39/54 (72.2)	28/52 (53.9)	0.050
Other	6 (1.6)	11 (3.3)									

Note: n = number of prescriptions for that indication that were compliant with or had appropriate drug choice/duration according to the Australian Therapeutic Guidelines. N = total number of prescriptions for that drug that were eligible to be assessed for compliance or appropriateness.

## Data Availability

The study datasets are not available due to participant privacy but can be available with a reasonable request from the corresponding author.

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
