# Peer review of "Is Education Alone Enough to Sustain Improvements of Antimicrobial Stewardship in General Practice in Australia? Results of an Intervention Follow-Up Study"

_antibiotics, 2023, doi:10.3390/antibiotics12030594_

Round 1
Reviewer 1 Report
Comments for authors
- Lines 26-29: Insert the p-values or any other statistical values to substantiate the significant reduction being reported
- Lines 29-30: If “simple and single-occasion antimicrobial stewardship education is probably not enough…” What should be done? (i.e your recommendation).
- Line 50 -52: The authors seem to be using GPs (earlier defined) and GP (not yet defined) interchangeably. Again, it’s not good to use the same word/acronym (GP/GPs) 3 times in the same sentence.
- Line 61: Replace “demonstrated” with “presented”
- Table 1 title: Insert “in Australia” after “demographics”. All table titles need to be fully descriptive and should be able to stand alone. Remove the full stop (.) after the last word of all your table titles
- Line 186-189: This should be listed as a major limitation to this study.
- Method: Since patient’s health data were obtained, did the authors request for and obtained informed consent from the patients or any of their authorised agent? Could the authors provide the appropriate version of the Australian Therapeutic Guideline used in this study for reproducibility of the work?
- Lines 263-266: Was there a form of quality control checks by another researcher (or was it a one time off thing) since compliance here can be subjective?
Author Response
Thanks so much for your valuable comments. All the recommended changes are found in track changes and responses are given below.
- Lines 26-29: Insert the p-values or any other statistical values to substantiate the significant reduction being reported- p value inserted
- Lines 29-30: If “simple and single-occasion antimicrobial stewardship education is probably not enough…” What should be done? (i.e your recommendation).- Described in the discussion section
- Line 50 -52: The authors seem to be using GPs (earlier defined) and GP (not yet defined) interchangeably. Again, it’s not good to use the same word/acronym (GP/GPs) 3 times in the same sentence.- "GP" has been spelled out.
- Line 61: Replace “demonstrated” with “presented”-amended
- Table 1 title: Insert “in Australia” after “demographics”. All table titles need to be fully descriptive and should be able to stand alone. Remove the full stop (.) after the last word of all your table titles- Amended
- Line 186-189: This should be listed as a major limitation to this study.-listed in the limitation section
- Method: Since patient’s health data were obtained, did the authors request for and obtained informed consent from the patients or any of their authorised agent? Consent has been provided by GP clinic manager on behalf of the GPs and which has been confirmed and approved by the ethics committee.
- Could the authors provide the appropriate version of the Australian Therapeutic Guideline used in this study for reproducibility of the work?- Version clarified
- Lines 263-266: Was there a form of quality control checks by another researcher (or was it a one time off thing) since compliance here can be subjective?- The assessment was not done by a single clinician but as a team of expert infectious disease physician and antimicrobial stewardship pharmacist. This is why no quality control checks was done by other researcher.
Reviewer 2 Report
Dear Editor;
Thank you for extending to me the privilege of reviewing this paper. The study under scrutiny examines the impact of education in antimicrobial stewardship programs, a subject of great interest and relevance in the realm of healthcare. Below, I offer my suggestions to further improve the quality of the paper.
It is my considered opinion that this study is well-suited to the scope of the journal and is likely to appeal to an international readership. The Journal of Antibiotics enjoys a prestigious reputation in academic circles, and its distinguished readership expects no less than exceptional and scholarly publications. In this regard, I believe that the current study meets the standards of quality necessary for publication in Antibiotics. The novelty of the study should be underline whether within the abstract or main text. There exist some minor issues that require attention and resolution.
Thank you for considering my review of this paper.
Comments
The title of the article under consideration is commendable as it effectively draws attention to the critical issue of antibiotic stewardship and the impact of education on it. The stated aim, target population, and study type are notably concise and substantial, contributing to the overall merit of the research. However, in the interest of enhancing readability, a more succinct rendition of the title that retains the same essence may be more appropriate for the intended audience.
The selection of appropriate keywords is crucial to ensure that the manuscript is effectively indexed and discoverable by readers. In this regard, it is imperative to adhere to the standardized Medical Subject Headings (MeSH) provided by relevant databases. By utilizing MeSH terms, authors can improve the visibility of their research, enhance search accuracy, and enable efficient retrieval of information. Therefore, I recommend that the authors carefully review the MeSH terms and choose those that are most relevant to the subject matter of their study. By doing so, they will facilitate the dissemination of their research to a broader audience and contribute to the advancement of knowledge in their field.
Regarding the content of the introduction section, it would be advantageous to provide a more explicit explication of the type of education utilized in the study. Based on my interpretation, it appears that the authors are referring to interprofessional education, but a more precise articulation would help to remove any potential ambiguity.
Based on my assessment, it is my opinion that the study under review adheres to the STROBE (Strengthening the Reporting of Observational Studies in Epidemiology) guidelines. To further enhance the comprehensiveness of the report, it would be beneficial to provide explicit details on the specific STROBE requirements that have been met. In addition, it is imperative that the required STROBE checklist is included as a supplementary file to the manuscript. As per the STROBE guidelines, it is vital to provide a detailed flowchart outlining the process of inclusion, randomization, and allocation in the study. This information should be presented in a clear and concise manner and in compliance with the STROBE requirements.
Based on my understanding, it appears that the selected techniques involve a mere comparison of the self-statements provided by the general practitioners. However, I am curious to know how the appropriateness of the prescribed antibiotics was determined.
Did the author make use of electronic health records to determine the appropriateness of the prescribed antibiotics? Alternatively, did they scrutinize the antibiogram results or merely verify that the antibiotics prescribed were suitable for the identified infections? Moreover, is it pertinent to consider the reliability of the chosen method in ascertaining the accuracy of the antibiotics prescribed.
Author Response
Thanks for your comments. Responses are as follows.
Comments
The title of the article under consideration is commendable as it effectively draws attention to the critical issue of antibiotic stewardship and the impact of education on it. The stated aim, target population, and study type are notably concise and substantial, contributing to the overall merit of the research. However, in the interest of enhancing readability, a more succinct rendition of the title that retains the same essence may be more appropriate for the intended audience.
Thanks for your comments and suggestion. We believe that the current title focuses education, sustainability of improvement, settings and study methods which are all important terms to attract readers.
The selection of appropriate keywords is crucial to ensure that the manuscript is effectively indexed and discoverable by readers. In this regard, it is imperative to adhere to the standardized Medical Subject Headings (MeSH) provided by relevant databases. By utilizing MeSH terms, authors can improve the visibility of their research, enhance search accuracy, and enable efficient retrieval of information. Therefore, I recommend that the authors carefully review the MeSH terms and choose those that are most relevant to the subject matter of their study. By doing so, they will facilitate the dissemination of their research to a broader audience and contribute to the advancement of knowledge in their field.
MeSH terms have been used.
Regarding the content of the introduction section, it would be advantageous to provide a more explicit explication of the type of education utilized in the study. Based on my interpretation, it appears that the authors are referring to interprofessional education, but a more precise articulation would help to remove any potential ambiguity.
Amended with inclusion of the following paragraph. “A one-hour face to face academic detailing session was implemented with GPs. Results of initial audits of antimicrobial prescription, guideline recommendations for common infections, regional antibiogram and pattern of resistance, AMS techniques such as delayed prescribing were shared. Infectious Diseases physician and AMS pharmacist facilitated the education session.” The detailed description exists in the method section.
Based on my assessment, it is my opinion that the study under review adheres to the STROBE (Strengthening the Reporting of Observational Studies in Epidemiology) guidelines. To further enhance the comprehensiveness of the report, it would be beneficial to provide explicit details on the specific STROBE requirements that have been met. In addition, it is imperative that the required STROBE checklist is included as a supplementary file to the manuscript. As per the STROBE guidelines, it is vital to provide a detailed flowchart outlining the process of inclusion, randomization, and allocation in the study. This information should be presented in a clear and concise manner and in compliance with the STROBE requirements.
The STROBE checklist has been used and attached as supplementary appendix 1.
Based on my understanding, it appears that the selected techniques involve a mere comparison of the self-statements provided by the general practitioners. However, I am curious to know how the appropriateness of the prescribed antibiotics was determined. Did the author make use of electronic health records to determine the appropriateness of the prescribed antibiotics? Alternatively, did they scrutinize the antibiogram results or merely verify that the antibiotics prescribed were suitable for the identified infections? Moreover, is it pertinent to consider the reliability of the chosen method in ascertaining the accuracy of the antibiotics prescribed.
The method section has described the assessment methods as follows
“A prior developed prescription assessment algorithm based on the Australian Therapeutic Guidelines: Antibiotic was used for assessment of guideline compliance and appropriateness. Infectious diseases physician (EA) and antimicrobial stewardship pharmacist (AN) assessed appropriateness and guideline compliance. A prescription was deemed compliant with the guideline when the choice, dosage, frequency and duration all aligned with the guideline. A prescription was considered appropriate, when the antibiotic choice, dosage, and duration were a reasonable selection for the indication provided based on a set of criteria that included a spectrum of activity, pharmacokinetics of the antibiotic, and the disease itself. Team based assessment was done to avoid any bias in the results.”
Reviewer 3 Report
The manuscript is well written, and the authors have presented their findings. The research question is relevant and the study design is appropriate for the objectives of the research. The manuscript provides new information on antibiotic education and has the potential to contribute significantly to the field. However, there is some that need to be addressed before this manuscript can be accepted for publication. In general, the manuscript could benefit from more detailed descriptions of interventions, a more explicit statement of the research question or hypothesis, more detail on the calculation of the sample size, and a more detailed discussion of the limitations of the study.
General Comments:
In the discussion section, the results of the study are looked at in-depth, with the most important findings highlighted and compared to other studies. The authors also suggest things that could be done in the future, such as education and feedback programs that can be tailored to certain antibiotics and infections. However, the discussion could be improved by addressing a few key issues.
1. The abstract gives a short summary of the study, but it is not clear about the interventions, the results, and the limitations. The abstract could be made better by emphasizing how new the study is and giving more information about the interventions, the main results, and the problems.
2. In the introduction, there is a clear explanation of what the study is about and why the research question was chosen. On the other hand, the research question or hypothesis could be stated more clearly in the introduction. The introduction could also use a more in-depth discussion of research on how long interventions that aim to improve antimicrobial stewardship in general practice are likely to last.
3. Methods: This section discusses how the study was set up, who participated, what they did, how the data was collected, and how it was analyzed statistically. But the methods section could use more information about how the sample size was calculated, who was included and who wasn't, and how the data was collected.
4. Results: The results section presents the findings of the study, but it could be improved by providing more detail on the statistical tests used and the effect sizes. The presentation of the data in a table could be improved by separating the data into subgroups and presenting the changes in the outcome measures as percentages. The results section could also benefit from a more detailed discussion of the limitations of the study.
5. The authors could provide a more detailed explanation of why guideline compliance and appropriateness of antimicrobial choice and duration declined in the follow-up period. While the discussion touches upon factors such as patient pressure and repeat prescriptions, it would be useful to explore these issues in greater depth and consider other potential reasons for the decline.
6. The authors could provide more information on the limitations of the study. For example, the study was conducted in a single Australian state, and the results may not be generalizable to other regions. The study also relied on self-reported prescribing data, which may be subject to bias or inaccuracies.
7. The discussion could be improved by offering more specific recommendations for future interventions. While the authors suggest that education and feedback programs can be targeted toward specific antibiotics and infections, they do not provide specific details on what these programs should entail or how they should be implemented. Providing more concrete recommendations would be useful to policymakers and healthcare professionals looking to improve antimicrobial prescribing practices.
8. The discussion section summarizes the study results and offers some useful insights into how future interventions can be targeted toward specific antibiotics and infections. However, by addressing the issues outlined above, the authors could provide a more comprehensive and actionable discussion that would be of greater value to readers.
9. The discussion section needs to be expanded. The authors should provide a more comprehensive discussion of the study results and relate these results to the existing literature. They should also discuss the limitations of the study and suggest directions for future research.
10. The references need to be reviewed and updated to include more recent studies.
Author Response
Comments and Suggestions for Authors
The manuscript is well written, and the authors have presented their findings. The research question is relevant and the study design is appropriate for the objectives of the research. The manuscript provides new information on antibiotic education and has the potential to contribute significantly to the field. However, there is some that need to be addressed before this manuscript can be accepted for publication. In general, the manuscript could benefit from more detailed descriptions of interventions, a more explicit statement of the research question or hypothesis, more detail on the calculation of the sample size, and a more detailed discussion of the limitations of the study.
General Comments:
In the discussion section, the results of the study are looked at in-depth, with the most important findings highlighted and compared to other studies. The authors also suggest things that could be done in the future, such as education and feedback programs that can be tailored to certain antibiotics and infections. However, the discussion could be improved by addressing a few key issues.
Thanks so much for your important comments to improve the quality of the paper. Responses are as follows.
- The abstract gives a short summary of the study, but it is not clear about the interventions, the results, and the limitations. The abstract could be made better by emphasizing how new the study is and giving more information about the interventions, the main results, and the problems.
Problems and intervention descriptions have been added.
- In the introduction, there is a clear explanation of what the study is about and why the research question was chosen. On the other hand, the research question or hypothesis could be stated more clearly in the introduction. The introduction could also use a more in-depth discussion of research on how long interventions that aim to improve antimicrobial stewardship in general practice are likely to last.
The introduction section has now added the explanation and information .
- Methods: This section discusses how the study was set up, who participated, what they did, how the data was collected, and how it was analyzed statistically. But the methods section could use more information about how the sample size was calculated, who was included and who wasn't, and how the data was collected.
This was an observational follow up study that collected one month data of all antimicrobial prescription to compare with one month data of the previous year. Statistical methods applied has been described in the method section.
- Results: The results section presents the findings of the study, but it could be improved by providing more detail on the statistical tests used and the effect sizes. The presentation of the data in a table could be improved by separating the data into subgroups and presenting the changes in the outcome measures as percentages. The results section could also benefit from a more detailed discussion of the limitations of the study.
Table 3 and 4 already presented the subgroup analysis results by type of antimicrobials and indications with valid statistical interpretations, percentage, and p values.
- The authors could provide a more detailed explanation of why guideline compliance and appropriateness of antimicrobial choice and duration declined in the follow-up period. While the discussion touches upon factors such as patient pressure and repeat prescriptions, it would be useful to explore these issues in greater depth and consider other potential reasons for the decline.
The reasons and factors have already been described in the discussion section.
“Failure to sustain or improve intervention effect can be multifactorial. As data was collected and analysed against the new eTG guideline “Antibiotic” version 16 – there may have been variation [15] in the guidelines that the GPs weren’t familiar with. The new guideline was only available for 3 months prior to data collection month, July 2019. This fact can be supported by findings that GPs in Australia prescribed a number of antibiotics that were ceased from the recommendation list of Therapeutic Guidelines [16]. In our post intervention study [8], GPs acknowledge that they had a lack of access to the online version of the guideline. The other factors which prevent GPs to follow optimal prescribing behaviours include patient pressure, patient expectation [17] and repeat prescriptions [18]. Repeat prescriptions are common in general practice in Australia [19] and often lead to an extended duration of therapy than guidelines [11]. No repeat prescription policy [20] was effective from 2020 thus this issue may contribute to the obtained results.”
- The authors could provide more information on the limitations of the study. For example, the study was conducted in a single Australian state, and the results may not be generalizable to other regions. The study also relied on self-reported prescribing data, which may be subject to bias or inaccuracies.
Amended.
- The discussion could be improved by offering more specific recommendations for future interventions. While the authors suggest that education and feedback programs can be targeted toward specific antibiotics and infections, they do not provide specific details on what these programs should entail or how they should be implemented. Providing more concrete recommendations would be useful to policymakers and healthcare professionals looking to improve antimicrobial prescribing practices.
Recommendations have been described in discussion section.
- The discussion section summarizes the study results and offers some useful insights into how future interventions can be targeted toward specific antibiotics and infections. However, by addressing the issues outlined above, the authors could provide a more comprehensive and actionable discussion that would be of greater value to readers. 9. The discussion section needs to be expanded. The authors should provide a more comprehensive discussion of the study results and relate these results to the existing literature. They should also discuss the limitations of the study and suggest directions for future research.
The discussion section has been expanded.
- The references need to be reviewed and updated to include more recent studies.
Two new references have been added currently 26 and 27.